# Intelligent Active Correction Technology and Application of Tower Displacement in Arch Bridge Cable Lifting Construction

**Nianchun Deng** [1], **Mengsheng Yu** [1,2,*] and **Xinyu Yao** [2]

1    College of Civil Engineering and Architecture, Guangxi University, Nanning 530004, China; dengnch@163.com
2    Guangxi Transportation Science and Technology Group Co., Ltd., Nanning 530007, China; xinyu1052007892@163.com
\*    Correspondence: xiaoluo19890316@163.com

**Featured Application: This method can be applied to high-precision control of long-span arch bridges when using oblique pull buckle hanging construction method.**

**Abstract:** In order to control the tower deviation in the cable hoisting process of long-span concrete-filled steel tubular arch bridge during construction monitoring, a practical method of tower deviation correction was studied and established. In the paper, based on the studies about the deviation error formation mechanism of tower in the process of cable lifting, the relevant formulas of the arch rib elevation changes caused by the change of tower state were deduced. The traditional control methods including increase of tower stiffness and strength of the anchor cable, are not effective, costly, and require a longer construction period. To overcome these defects, in virtue of the Beidou GNSS measurement system and hydraulic jack active control system, the active control technology of the CFST (Concrete Filled Steel Tube) arch bridge tower deviation was thoroughly studied. Besides, a perfect active control theory was established. Finally, the author puts forward the idea of reverse pulling tower deviation. The field measurement and comparative study show that after the optimization of this method, the tower deviation is within 2 cm, and the error meets the specification requirements. The proposed method can accurately control the tower deviation in the process of arch bridge cable hoisting, and establish a set of perfect active control related systems and theories, which is especially suitable for the construction monitoring and adjustment in the construction stage of arch bridge, and can provide reference for the construction control of tower deviation of the same type of bridge.

**Keywords:** CFST arch bridge; Beidou GNSS; cable lifting; tower top displacement; automatic correction

## 1. Introduction

Compared with other types of bridges, the CFST (Concrete Filled Steel Tube) arch bridge has a higher bearing capacity, a better plasticity and strength, and is costly and convenient during construction. It has also solved the application and construction problems of high strength materials for arch bridge with long span [1–6]. With Chinese characteristics, it has certain advantages and broad application prospects in the future medium and long span bridges [7]. Due to its wide range of application, especially for long span bridges, cable lifting construction has become the main construction method of arch bridges [8–11]. In order to simplify and speed up the construction, and save costs, the tower bottom consolidation, tower and buckle frame integration structure was adopted [12]. When using cables to lift arch ribs, the working procedure of each span influences the deformation of the tower, causing deviation, and has a linkage effect on the arch rib hanging on it. When the linkage effect accumulates gradually, the assembly precision and arch rib linear will be affected if not adjusted [13–15], with the stress increased and the stability of tower reduced. The traditional passive control on a tower's horizontal deviation is usually achieved by rigidity,

which often increases the section of wind cables and wastes materials. Therefore, relevant experts and scholars put forward the automatic dynamic adjustment technology of cable fastening [16]. In the late 1980s, some researchers began to study digital tensioning [17], and now, intelligent prestressed tensioning technology can be realized [18], and its accuracy is higher than that of traditional tensioning equipment [19]. There is much equipment in the traditional hoisting system, which is difficult to adjust and construct [20], thus the prestressed intelligent tensioning system is applied to the bridge [21]. In the construction of Guangxi Yongjiang River Bridge, the method of jacking and cable-stay suspension is studied and applied for the first time [22]. However, it is not used for bridge monitoring. Table 1 below lists the latest research results on intelligent tower deviation control systems and analysis and measurement methods.

**Table 1.** The studies about smart tower deviation control systems and the methods of analysing and measurement.

| Research Content | Studies | Research Flaws |
|---|---|---|
| Smart tower deviation control systems Smart tower deviation control systems | In 2020, Feng [23] introduced a GNSS-based active control technology for the longitudinal displacement of the tower top. <br><br> In 2020, Pan [24] used the intelligent active load regulation technology to adjust the arch rib deformation during the concrete-filled steel tube pouring process | Through theoretical analysis, the feasibility of the smart tower deviation control system was verified, but no actual engineering application was carried out. Pan [24] used intelligent active load regulation technology to control the deformation of the arch rib during the pouring process, but did not consider the deformation of the tower. |
| The methods of analysing and measurement | Since 2015, Yu [25] and He et al. [26] studied the application of Beidou/GPS high-precision monitoring in bridge engineering, showing that Beidou has a stable and reliable dynamic monitoring. <br> Wang J. and Wang L. [27] conducted a precision calibration test on the Beidou monitoring system, and the results show that the horizontal displacement measurement accuracy is within 2 mm, and the settlement measurement accuracy is within 5 mm, proving that the accuracy of automatic satellite monitoring is consistent with the manual monitoring of total station, and it is not easily affected by the environment. | Failure to apply satellite monitoring methods to the bridge. <br><br><br> It proves that the satellite monitoring system replaces manual measurement with higher accuracy and can meet construction needs, but it is not used for bridge monitoring. |

This article mainly has the following novelties:

1. In virtue of the Beidou GNSS measurement system and hydraulic jack active control system that have been developed rapidly in recent years, the active control technology of the concrete-filled steel tubular arch bridge tower deviation was thoroughly studied. Besides, a perfect active control theory was established.
2. The paper designs and puts forward the smart active tower deviation control system to active control the horizontal displacement of the tower.
3. Through the measuring instruments and smart equipment with high precision, the paper solves the tower deviation in the process of traditional cable lifting system, and reduces the impact on the precision of arch truss cantilever.

Thus, it is a scientific, feasible, economic, and reasonable optimization construction scheme.

## 2. Analytical Model and Equations

### 2.1. Mechanism of Tower Deviation Error Formation

Construction control mainly aims to ensure that the final construction meets the design requirements, and the error is why results deviate from the design state. Many factors can cause the structural error of arch ribs, among which tower deviation and cable force

deviation are two important factors. To study the impact of tower deviation, the cable force is fixed, only the influence of the tower deviation on the feed rib is studied. The stress deformation of the tower will lead to the linkage effect of the arch ribs on the suspension and affect the suspension precision of the arch girder. In order to explore the basic principle of tower deflection, the following two assumptions are made: (1) assuming that the total length of buckle cable and anchor cable remains unchanged, it is considered that the cable will not be adjusted after the buckle cable is tensioned; (2) It is assumed that the buckle tower and arch rib segments are rigid bodies, there is no elastic deformation, only rigid body displacement.

In Figure 1, $h_1$ is the height of the tower; $L_b$, $L_s$ are anchor cable and buckle cable lengths before the tower deviates, respectively; $L_b'$, $L_s'$ are the anchor cable and buckle cable lengths after the tower deviates; $h_2$, $h_2'$ are the distance between the top of the tower and the center of the arch foot before and after the deviation; S is the chord length of the arch rib section; is the angle between the chord length of the front arch rib and the horizontal line. In accordance with the geometric relationship, $\theta$ is the angle between the chord length of the front arch rib and the horizontal line. According to the simplified model, the relationship between tower deflection and arch rib error can be established as follows:

$$\cos \alpha_1 = \frac{a_1{}^2 + h_1{}^2 - L_b{}^2}{2a_1 h_1}, \tag{1}$$

$$\beta = \arctan(\frac{\delta}{h_1}) \tag{2}$$

$$L_b' = \sqrt{a_1{}^2 + h_1{}^2 - 2a_1 h_1 \cos(\alpha_1 - \beta)} \tag{3}$$

$$\cos \alpha_2 = \frac{a_2{}^2 + h_1{}^2 - h_2{}^2}{2a_2 h_1} \tag{4}$$

$$h_2' = \sqrt{a_2{}^2 + h_1{}^2 - 2a_1 h_1 \cos(\alpha_2 + \beta)} \tag{5}$$

$$\cos \gamma = \frac{h_2{}^2 + (h_2)'^2 - \delta}{2h_2 h_1} \tag{6}$$

$$L_s' = L_b + L_s - L_b' \tag{7}$$

$$\cos \phi = \frac{h_2{}^2 + S^2 - L_b{}^2}{2h_2 S} \tag{8}$$

$$\cos(\gamma + \phi + \Delta\phi) = \frac{h_2{}^2 + S^2 - L_S{}^2}{2h_2' S} \tag{9}$$

Thus, the vertical displacement from $A$ to $A'$ point is: $\Delta_y = S \sin\theta - S \sin(\theta - \Delta\phi) = S \sin\theta - S \sin\theta \cos\Delta\phi + S \cos\theta \sin\Delta\phi \approx S \sin\theta - S \sin\theta + S\Delta\phi \cos\theta$, where $\cos\Delta\phi \approx 1$; $\sin\Delta\phi \approx \Delta\phi$. Equations (1)–(9) are the influence of deviation of buttress tower on the elevation of arch rib section, where the buttress tower has elastic deformation under unbalanced force. The elevation of control points is higher than the predicted linear shape under the impact of tower deviation, which is consistent with the measured data. Therefore, the impact of tower structure should be considered in construction control to guarantee the safety of bridge during construction.

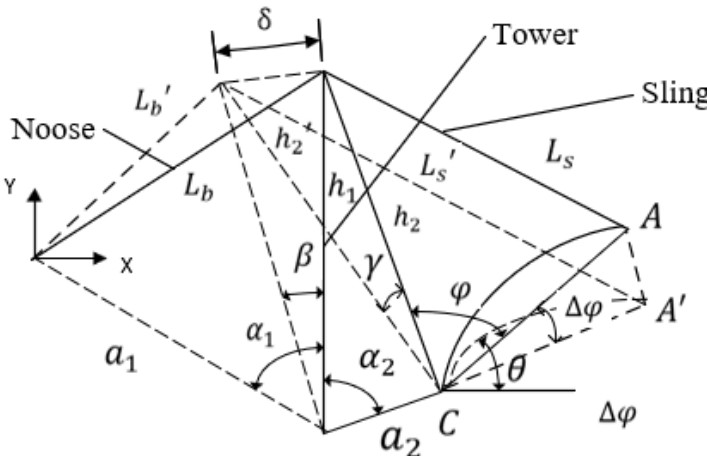

**Figure 1.** The impact of tower deviation on segment elevation.

### 2.2. Passive Control Theory

The displacement of the tower top is affected by the lifting weight of the cable, and increases with the increase of span and section weight. The tower top displacement control also becomes more difficult. Traditionally, the manual operation of jacks is used as the passive control method to correct tower deflection. In fact, the tower deflection adjustment method is inefficient. Furthermore, with the traditional method, it is difficult to ensure that the load adjustment effect achieves the ideal state, and it is time-consuming and laborious, especially when the span of the arch rib increases, the corresponding number of arch rib segments increases, and each lifting segment needs to control the deviation of the tower top under a reasonable target state, which requires constant adjustment. Thereby, it is difficult for the existing adjustment method of tower top deviation to meet the construction control requirements.

In the passive control proposed by Zheng et al. [28], the constantly changing horizontal force of the main cable on the top of the tower during the lifting, the transportation and suspension of the arch rib section is $F(t)$, and the horizontal displacement generated by the top of the tower is $H(t)$. Assuming that the section of wind cable is section $A$, the elastic modulus is $E$, the resistance level of push–pull stiffness is $K$, the horizontal tensile stiffness coefficient of wind cable is $r_i$, then:

$$H(t) = \frac{F(t)}{K + \sum i r_i A_i / L_i} \tag{10}$$

As seen from the formula, the horizontal displacement of the tower is determined by the horizontal bending stiffness of the tower and the area of the wind cable, while the strength of the wind cable is far from being exerted.

### 2.3. Theoretical Verification of Automatic Correction

Through the introduction of the main control in the previous section, this section will further prove the feasibility of active control. Considering that the sum of the deformation potential energy of the structural element is equal to the work done by the external force acting on the structure, the following formula can be obtained:

$$\delta U_{Total} + \delta W = 0 \tag{11}$$

The force on the structure is deduced by the variational method:

$$\sum_{j=1}^{n} D\delta = \{P\} + \sum_{j=1}^{n} L_0, \tag{12}$$

$$L_0 = \left\{ \begin{array}{c} (l - l_0)\frac{EA}{l}c + \left(K_{io} - K_{jo}\right)\frac{EA}{l}s \\ (l - l_0)\frac{EA}{l}s - \left(K_{io} - K_{jo}\right)\frac{EI}{l}c \\ -K_{jo}EI \\ -(l - l_0)\frac{EA}{l}c - \left(K_{io} - K_{jo}\right)\frac{EI}{l}s \\ -(l - l_0)\frac{EA}{l}s + \left(K_{io} - K_{jo}\right)\frac{EI}{l}s \\ K_{jo}EI \end{array} \right\} \tag{13}$$

In the above formula: $\delta U_{Total}$ is the potential energy of structural deformation; $\delta W$ is the potential energy of structural deformation; $K_{io}$, $K_{jo}$ are curvature of member $i$ and $j$; $E$, $A$, $l$ are the elastic modulus, cross-sectional area, length of the rod; $L_0$ is the generalized additional load.

According to Equation (13) above, it can be seen that the bridge structure state that determines the phased construction is mainly related to boundary conditions, external load and structural system. Thereby, the structure state can be changed by changing any of the conditions, but it is difficult to change the structure system and boundary conditions of the tower that have been installed and constructed. Moreover, the target state of the tower can be adjusted by changing the external load [29]. According to the general engineering practice research, tower deflection is mainly caused by the horizontal force in the longitudinal direction of the bridge. The vertical force in the resultant force of the cable system is small relatively to the dead weight of the tower, thus the control of horizontal force is the key to tower deviation correction. Under the impact of the lifting weight, the tower is affected by the horizontal force P in the direction of the river. The stress state of the tower with only horizontal force during the lifting process of the arch rib is established, as shown in Figure 2.

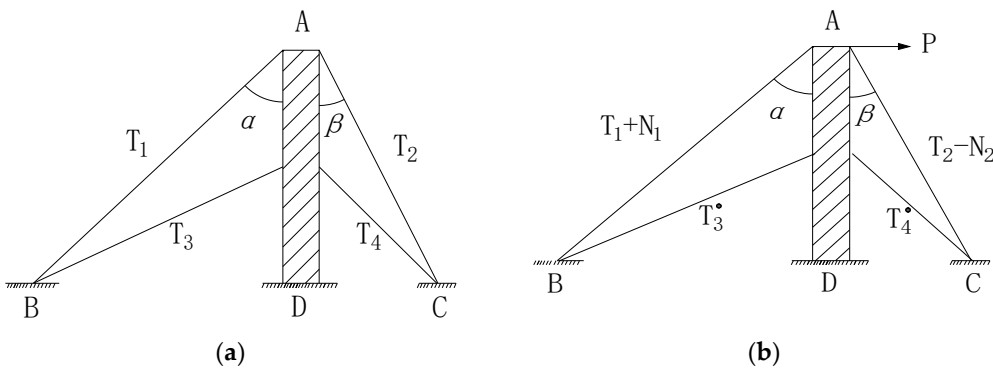

**Figure 2.** The stress state of wind cable cantilever beam under horizontal force only. (**a**) Initial stress state; (**b**) Force state under horizontal force.

The initial state is shown in Figure 2a. The B end is the bank, and the C end is the middle of the river. The A end of the cantilever beam is constrained by the wind cable AB and AC. The inclined angle is assumed to be $\alpha$ and $\beta$, and the pretension to be $T_1$ and $T_2$, respectively. When end A is subjected to horizontal force P (Figure 2b), the end A has a tendency to move to the right to make section AB stretch and load, the size is $N_1$ and end A has a tendency to move to the left to shorten section AC and unload, and the size is $N_2$. At this point, the internal forces of the wind cables on both sides are $(T_1 + N_1)$ and $(T_2 - N_2)$, respectively.

$$T_1 \sin \alpha = T_2 \sin \beta, \tag{14}$$

Assuming that the shear force at the A end is to the right, with the size of $Q$, then the system re-reaches balance after active regulation. Based on the balance condition $\sum F_X = 0$, the following relation can be reached:

$$(T_1 + N_1) \sin \alpha = Q + (T_2 - N_2) \sin \beta + P, \tag{15}$$

Assuming that the regulating horizontal force is $Q_0$, after substituting Equation (14) into Equation (15), the top shear force is:

$$Q = (Q_0 + N_1 \sin\alpha + N_2 \sin\beta) - P, \tag{16}$$

The top shear force is discussed as follows:

(1)  When $Q_0 = 0$, $N_1 = N_2 = 0$, i.e., it is in the state of no pretension. According to Equation (16), $Q = -P$, indicating that the direction of $Q$ is left.

(2)  When $Q_0 \neq 0$, $N_1 \neq 0$, $N_2 \neq 0$, the corresponding length of cantilever beam is $L_0$. To make it satisfy the Equation (16), $Q = 0$, indicating that the state is the critical state for the change of $Q$ direction.

(3)  When $Q_0 \neq 0$, $N_1 \neq 0$, $N_2 \neq 0$, if its corresponding cantilever beam length $L > L_0$, it shows that the cantilever beam line stiffness decreases, resulting in the increased deformation, or obvious load and unload effect; let $Q_0 + N_1 \sin\alpha + N_2 \sin\beta > P$, according to Equation (16), $Q > 0$, showing that the direction of the $Q$ is right.

(4)  When $Q_0 \neq 0$, $N_1 \neq 0$, $N_2 \neq 0$, if its corresponding cantilever beam length $L < L_0$, showing that the stiffness of the beam line increases, leading to the reduced deformation, thus reducing the unloading effect; make $Q_0 + N_1 \sin\alpha + N_2 \sin\beta < P$, according to Equation (16), $Q < 0$, showing that the direction of the $Q$ is left.

Based on the above analysis, when the cantilever beam is under the horizontal load $P$, the direction of the top shear force is mainly related to the initial tension of the structure and the linear stiffness of the cantilever beam. When the initial tension or the structure line stiffness changes, all of the above four situations may occur. Accordingly, only when the direction of $Q$ changes to the critical state, the horizontal impact on tower deviation can be eliminated. By comparison, the active control method can achieve the real-time dynamic adjustment of tower deviation to the target state, making tower under the condition of changing the horizontal force reach a critical state through the feedback control system, eventually eliminates the impact of horizontal force, and keeps the structure in vertical state. Therefore, the feasibility of the active control is theoretically proved.

## 3. Proposed Methodology of Monitoring

### 3.1. Intelligent Active Deviation Correction System

The system is mainly composed both of Beidou GNSS real-time data monitoring branch system and the feedback control branch system. The monitoring system tracks GNSS satellite and collects data in real time. Moreover, the deformation of the tower is monitored and sent to the feedback control system in real time. The GNSS displacement measurement system of Beidou adopts the national geodetic coordinate system CGCS-2000, which takes east, west, north and south as the coordinate axis. The tower displacement measurement focuses on the deviation of the central axis of the tower on the north bank south.

With Matan Hongshui He Bridge as the engineering background, the two Beidou measuring points of the tower integrating with main cable and buckle are located in two sides of the transverse bridge. The overall layout of the lifting construction system is shown in Figure 3 below. The space deformation of cable tower (such as reverse) is measured.

### 3.2. Tower Active Control System Components

As shown in Figure 4a above, Beidou users receive the navigation displacement monitoring signal from the Beidou satellite through the Beidou satellite GNSS antenna installed on the tower and the system autonomously and accurately calculates 3D pose parameters (X, Y and Z) using the N72 reference receiver (Figure 4b) installed on the tower. Then, the data is transmitted to the base station through the bridge and processed and analyzed by the relevant software in the workstation. The data are sent to the workstation through wireless bridge switches. The base station and the specific working principle are shown in Figure 5.

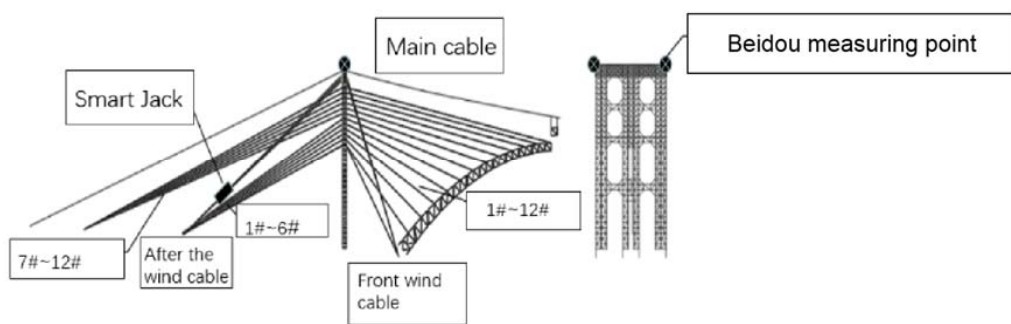

**Figure 3.** Overall layout of lifting construction system.

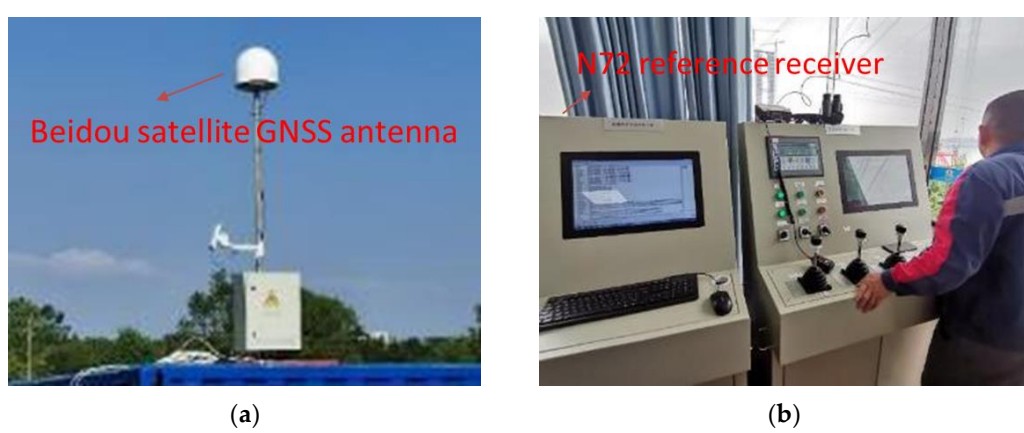

**Figure 4.** Beidou GNSS satellite measurement system and supporting facilities. (**a**) Control room; (**b**) Beidou satellite station and demodulator.

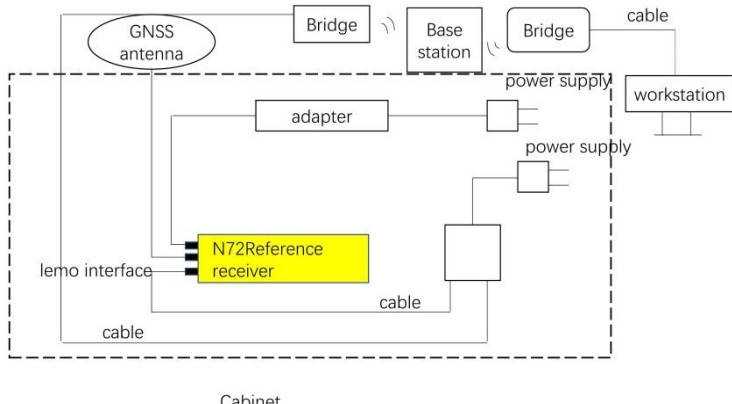

**Figure 5.** The working principle of the data monitoring system.

### 3.3. The Feedback Control System

Moreover, the feedback control system uses the three-dimensional mm level displacement data of Beidou GNSS real-time monitoring system. Based on the feedback of the automatically-measured data, the closed loop control of smart pump station and smart jack tensioning is thus achieved through controlling the horizontal displacement of the tower to the target state.

The feedback control system in the paper mainly refers to the smart hydraulic jack tension system. It can automatically control the tension and release of the smart pump station and the smart jack through the control data received and processed by the workstation, as shown in Figure 6 below.

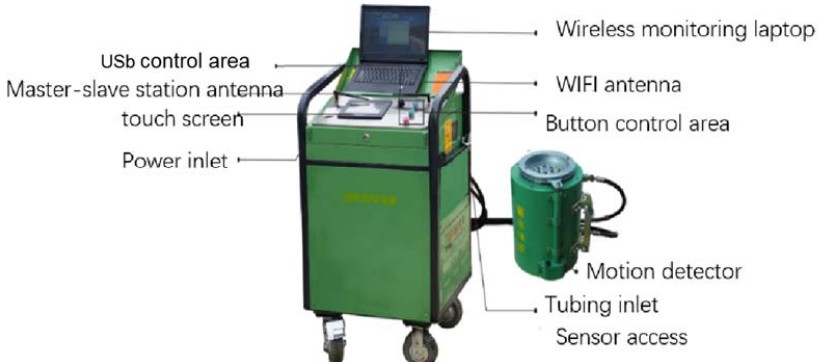

**Figure 6.** The feedback control system and supporting facilities.

The feedback control system collects data based on Beidou data monitoring. The software automatically issues instructions, and realizes precise tension of wind cable and cable according to the feedback of the automatic measurement data of the displacement sensor on the smart pump, making the tower keep numerical state. In addition, it stores the data in the process of tension, automatically generates reports, and shows the historical data of tension at any time. Thus, it can eliminate the interference of human factors and effectively guarantee the quality of tensioning construction. The working principle is shown in Figure 7. Intelligent active control is to install the intelligent Jack after the initial tensioning of the designed active control cable wind, monitor the structural deviation through the GNSS automatic displacement acquisition system, judge the structural displacement by the controller, and send the command of Jack tensioning or retraction to the intelligent hydraulic pump station to realize the tensioning and retraction of cable force, so as to reduce the structural displacement. Figure 8 below shows the schematic diagram of top displacement smart active control.

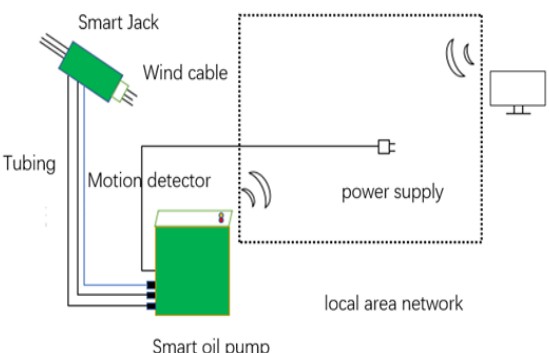

**Figure 7.** The working principle of smart tension system of hydraulic Jack.

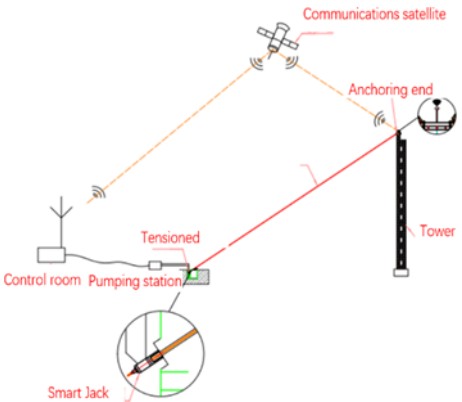

**Figure 8.** Schematic diagram of tower top displacement control.

## 4. Proposed Methodology of Monitoring

### 4.1. Finite Element Modeling

In this section, the influence of intelligent active control on the stress, deformation, and stability of the tower is simulated by the finite element software MIDAS/Civil. The three-dimensional structure tower model is established, and the half-span is analyzed to model the suspended arch rib of Matan Hongshui He Arch Bridge and Pingnan Third Bridge [30] (as seen in Figure 9).

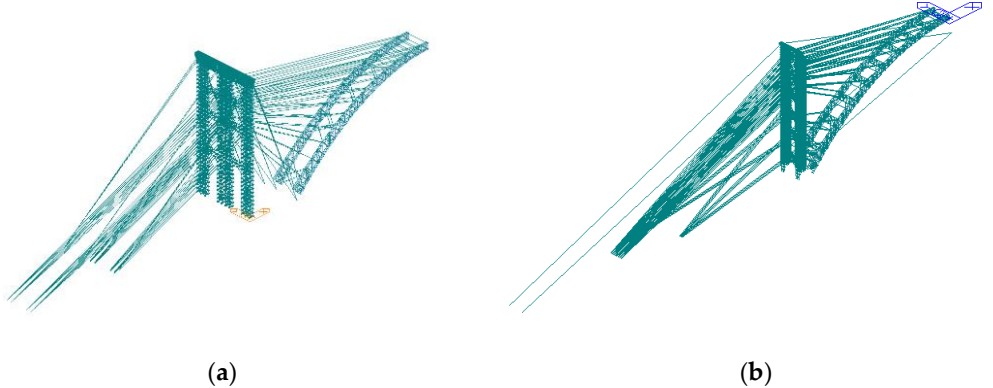

(**a**)                                                              (**b**)

**Figure 9.** Model diagrams of Matan Hongshui He Bridge and Pingnan Third Bridge. (**a**) Model diagram of Matan Hongshui He Bridge; (**b**) Model diagram of Pingnan Third Bridge.

Tower structure is simulated according to the actual construction sequence. The wind cable is simulated by the cable element, and the stiffness reduction is also considered [31]. The rest is simulated by beam element, and there are 6756 nodes and 13,678 elements in this model. Since the tower bottom is build-in on bedrock, the tower is fixed in this model. Based on the lifting weight, the impact effect and the influence of small wheel pressure are considered. Besides, the wind load is six degree (from the river to the bank direction) under normal working conditions. The arch rib is constructed by cantilever construction with cable-stayed button-hanging method, and it includes 12 installation sections. The mass of the arch rib will produce force on the button-hanging point of the tower through the cable when the arch rib is button-hanging. As for the initial tension of the wind cable, the tower is connected with the cable by nodal plates and bolts. Therefore, during modeling, the actual dead weight of the tower is multiplied by a coefficient of 1.2 times before it is input into the model for calculation. In a cable-arch bridge or in a cable-stayed bridge, the flexural rigidity of slender cables can generally be neglected. Conversely, it cannot be negligible for the shortest ones. The accuracy of cable force calculation is directly related to the boundary conditions and calculation model at both ends of the cable. Stiffness and boundary conditions are the two most important factors in the establishment of short cable model. The accurate measurement method of cable force is adopted [32], which comprehensively considers the influence of bending stiffness and boundary conditions. At the same time, the bending stiffness is taken as an implicit calculation parameter, which avoids the problem that it is difficult to identify accurately, reduces the measurement difficulty, and improves the measurement accuracy [33–35].

The cable lifting load and buckle load are equivalent to node load, and the wind load is equivalent to surface load. The initial cable tension and active control force are applied to the wind cable element as the initial tension load. Active control only occurs in the working state. In the paper, model for the normal working state and the active regulation under normal working state is established respectively. Because load combination 1 is the most unfavorable working condition under the working status, the most unfavorable condition is actively adjusted to load combination 2. The five types of load combination effect are shown in Table 2 below. Moreover, the static characteristics are studied under each load combination.

**Table 2.** Various working conditions of construction.

| Project | Number | Conditions | Load Combination |
|---|---|---|---|
| Matan Hongshui He Bridge | Condition 1 | Normal hanging arch ribs | Cable action, self-weighting, buckle load, 6 levels of vertical wind, cable air rope initial tension |
| | Condition 2 | Normal hanging arch ribs under the action of active control force | Cable action, self-weighting, buckle load, 6 levels of vertical wind, cable initial tension, active control force |
| Pingnan Third Bridge | Condition 1 | Normal hanging arch ribs | Cable action, self-weighting, buckle load, 6 levels of vertical wind, cable air rope initial tension |
| | Condition 2 | Normal hanging arch ribs under the action of active control force | Cable action, self-weighting, buckle load, 6 levels of vertical wind, cable initial tension, active control force |

### 4.2. Deflection Displacement Analysis

The horizontal displacement of the tower is very small as it is controlled by the lateral wind cable, thus it is ignored in the paper. The maximum horizontal and vertical displacement of the newly-built tower is occurred at the top of the tower. The results of the displacement at the tower top are shown in Table 3 below.

**Table 3.** The calculated results of tower top displacement (mm).

| Project | Direction | Condition 1 | Condition 2 | Regulation Rate of Change (%) |
|---|---|---|---|---|
| Matan Hongshui He Bridge | Horizontal (mm) | 153.00 | 18.10 | −88.17 |
| | Vertical (mm) | −48.80 | −51.08 | 4.67 |
| Pingnan Third Bridge | Horizontal (mm) | 185.22 | 17.07 | 90.78 |
| | Vertical d(mm) | −83.39 | −78.78 | 5.50 |

According to the results, the active control has a significant influence on the horizontal displacement of the tower, which ranges from 80% to 90%. However, the influence on the vertical displacement ranges from 4% to 5%, which is much smaller compared to the influence of the horizontal displacement. As a whole, the tower deformation of Matan Hongshui He Bridge is dominated by horizontal deformation, as the vertical deformation is very small. Under the active control, the tower deviations of Matan Hongshui He Bridge and Pingnan Third Bridge are 18.1 mm and 17.07 mm, respectively. Thus, it can be concluded that the major effect of lateral load can be eliminated by active control, which proves theoretically the feasibility of the active control.

### 4.3. Stress and Internal Force Analysis

Stress and internal force are the most important indexes to investigate the tower bearing capacity. Table 4 below shows the internal force and stress values of key sections under various working conditions, and presents a comparison between working conditions.

**Table 4.** Stress and internal force under various working conditions.

| Project | Result | Condition 1 | Condition 2 | Regulation Rate of Change (%) |
|---|---|---|---|---|
| Matan Hongshui He Bridge | Column steel pipe maximum stress (MPa) | 118.90 | 117.40 | 1.28 |
| | Tower top structure Maximum stress (MPa) | 122.10 | 122.70 | 0.49 |
| | Buckle anchoring maximum stress (MPa) | 88.60 | 89.50 | 1.01 |
| | Tower top shaft force (kN) | 219.70 | 337.80 | 34.96 |
| | Base of the tower shaft force (kN) | 2679.50 | 2870.10 | 6.64 |
| | Tower top bending moment (kN·m) | 290.72 | 282.90 | 2.76 |
| | Bottom of the tower bending moment (kN·m) | 63.90 | 62.90 | 1.59 |
| Pingnan Third Bridge | Maximum stress of column steel pipe (MPa) | 182.10 | 193.9 | 6.48 |
| | Maximum stress of tower structure (MPa) | 60.20 | 61.9 | 2.82 |
| | Maximum stress of tower web (MPa) | 62.11 | 64.00 | 3.04 |
| | Internal force at the bottom of the tower (kN) | 3679.20 | 3769.10 | 2.44 |

As the results show, the maximum stress for the tower of Matan Hongshui He Bridge in working condition 1 and 2 is 122.1 MPa and 122.7 MPa; the maximum stress for the tower of Pingnan Third Bridge in working condition 1 and 2 is 182.1 MPa and 193.9 MPa. As a whole, the effect of active control on the stress of the structure is not obvious. It is clear that the active control has a smaller impact on the stress of the structure. The active control force exerts obvious impact on the horizontal displacement of the tower, but it has very low sensitivity to the stress and internal force of the tower. According to the strength design value of CFST tower and the steel content ratio of the bridge tower, the maximum limit value is calculated to be 203 MPa, which meets the requirements of the specification and shows the tower is in a safe state.

### 4.4. Stability Analysis of Tower under Smart Active Control

Table 5 below lists the first five critical load coefficients of various working condition. Figure 10 below compares the critical load coefficients of various working condition.

**Table 5.** The Stability coefficient under various working conditions.

| Project | Instability Order | Condition 1 Critical Load Factor | Condition 2 Critical Load Factor | Regulatory Change Rate (%) |
|---|---|---|---|---|
| Matan Hongshui He Bridge | 1 | 24.15 | 23.32 | 3.44 |
| | 2 | 30.53 | 29.44 | 3.57 |
| | 3 | 41.76 | 37.81 | 9.46 |
| | 4 | 45.77 | 45.47 | 0.66 |
| | 5 | 45.77 | 40.33 | 11.89 |
| Pingnan Third Bridge | 1 | 25.42 | 27.54 | 8.34 |
| | 2 | 30.40 | 29.11 | 4.24 |
| | 3 | 39.12 | 38.17 | 2.43 |
| | 4 | 45.41 | 46.33 | 2.03 |
| | 5 | 47.54 | 46.71 | 1.75 |

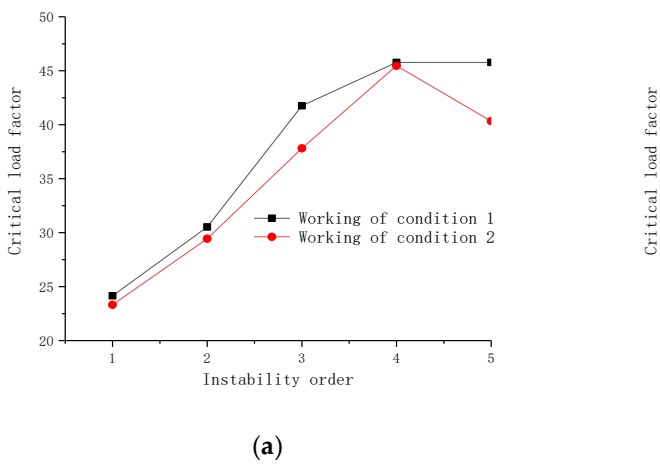 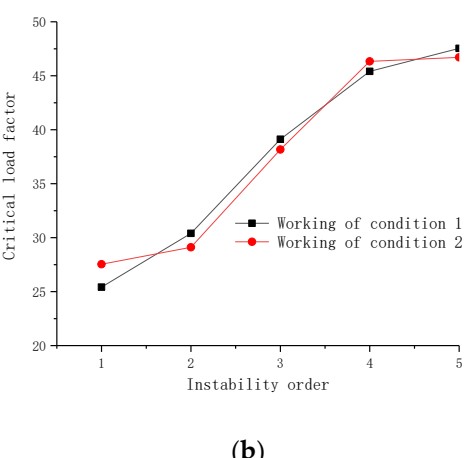

(a) (b)

**Figure 10.** Comparison of critical Stability coefficients under various working conditions. (**a**) Comparison of critical Stability coefficients in Matan Hongshui He Bridge; (**b**) Comparison of critical load coefficient in Pingnan Third Bridge.

As the results show, the first-order critical Stability coefficient for the tower of Matan Hongshui He Bridge in working condition 1 and 2 is 24.15 and 23.32; the first-order critical Stability coefficient for the tower of Pingnan Third Bridge in working condition 1 and 2 is 25.42 and 27.54. With the application of active load under normal working conditions, the stability coefficient of the structure decreases slightly, and the impact of active control on the first-order critical load coefficient is 3~4%. From the aspect of instability morphology, there are vertical vibration and torsional vibration of the tower under most of the working

conditions. Moreover, with the increase of instability order, the stability coefficient increases gradually, that is to say, higher-order instability becomes progressively less possible.

## 5. Engineering Application of Smart Active Control

### 5.1. Engineering Background

The main bridge of the Guangxi Laibin Matan Hongshui He Extra Large Bridge adopts a half-through CFST arch bridge with a span of 336 m, which is divided into two left and right sections. A single arch rib is divided into 24 sections, and the whole bridge has a total of 96 sections. The maximum weight of the single arch is 1150 KN and the minimum weight is 541 KN. The rise–span ratio is 1/4, and the inverted catenary arch axis coefficient is 1.167, and a variable-height steel tube concrete truss structure is used. The arch foot truss is 12 m high, and the arch top truss is 7 m high with a width of 3 m. The tower system adopts the method of "hanging and buckling integrated", i.e., the hanging tower and the buckling tower are integrated. There are two sets of main cables, with a rated lifting force of 1000 KN. The cable lifting system tower is set up by large steel pipes, and the total height of the tower is 130 m, Figure 11 is the schematic diagram of the half-span structure of the main bridge, and Figure 12 is the schematic diagram of the tower layout.

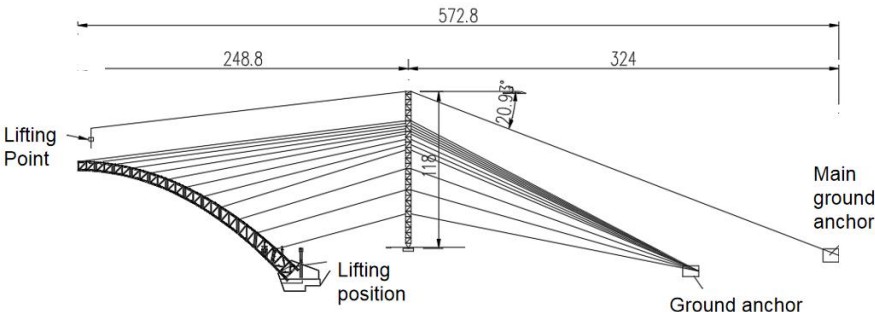

**Figure 11.** Schematic diagram of the half-span structure of the main bridge (m).

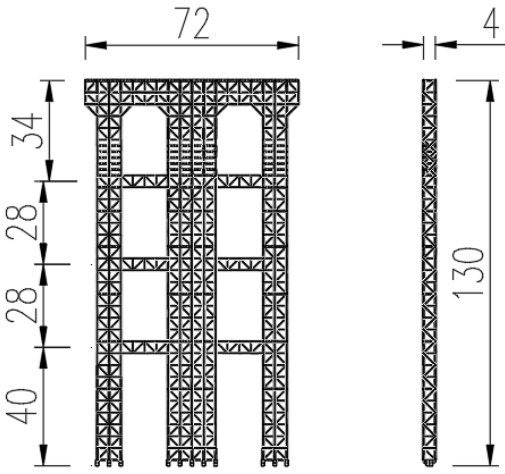

**Figure 12.** Schematic diagram of tower layout (m).

In particular, Pingnan Third Bridge (Figure 13), the world's first arch span of 575 m, also adopts this research technology.

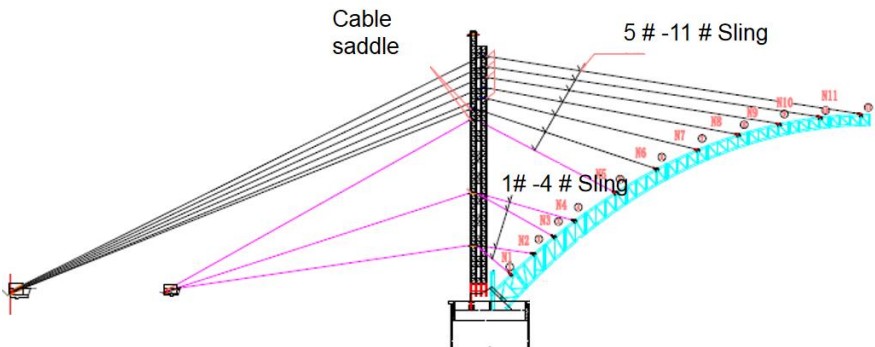

**Figure 13.** Pingnan Third Bridge layout drawing of half-span anchorage system.

This section describes in detail the application of the technology in the Matan Hongshui He Bridge. Two Beidou measuring points are arranged on the top of the tower with the main cable and buckle in one, which are located on both sides of the horizontal bridge, and they can measure the spatial deformation of the cable tower in real time (Figure 14). Load adjustment is conducted in the process of lifting arch rib. Before lifting, the active control system is started, which is similar to the only prestress tensioning condition. After the load adjustment is in place, and the arch rib lifting is finished, the suspension centre is released and the system is closed. Combined with the on-site cable crane system, the horizontal displacement system of the smart active control (Figure 15) button tower uses the original wind cable for active control when the arch ribs are lifted. The equipment parameters and target state control parameters are determined in advance. To make the tower keep upright in the process of lifting arch rib and transportation all the time, it is necessary to constantly feedback the closed loop control. Figure 16 is the schematic diagram of jack pulling wind cable in process of the active control of tower.

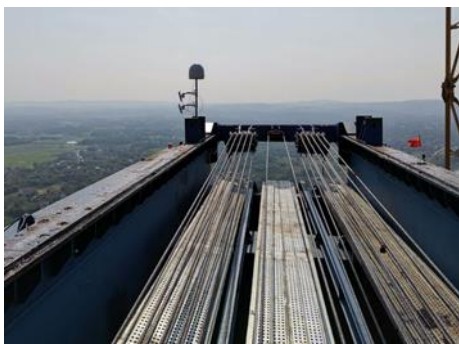

**Figure 14.** Schematic diagram of cable tower Beidou measuring point arrangement.

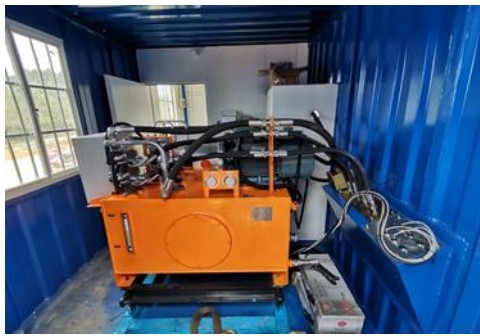

**Figure 15.** Workstation intelligent control hydraulic pump.

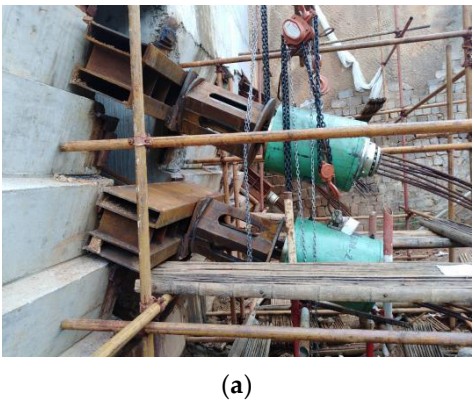 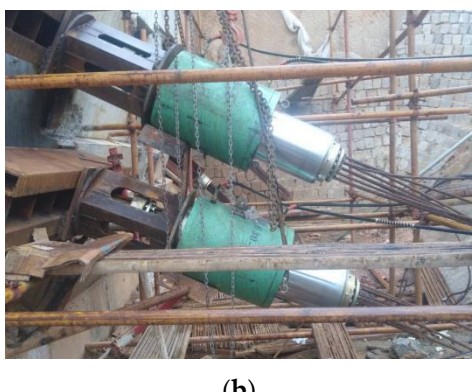

(**a**) (**b**)

**Figure 16.** The schematic diagram of active control. (**a**) Jack tension initial state; (**b**) Jack tension state.

### 5.2. Active Control Process and Data Analysis of the Tower

The GNSS displacement automatic monitoring system dynamically calculates the observation data to obtain the single period coordinates of each monitoring point, and intuitively obtain the instantaneous changes of the off-balance along the bridge of the tower top in different lifting states. Due to the space limitation, only partial results are selected. The construction group division and monitoring lifting sections are shown in Table 6 below. To eliminate the impact of high frequency noise in the monitoring results, the single period positioning results of the measuring points are also filtered [36]. The segments at both sides of the bridge are the same, thereby, the south and north banks can be used as the control group. Figure 17 below shows the monitoring results of GNSS in each construction group.

**Table 6.** The experiment construction group division.

| Construction Group 1 | Unregulated Segment (South Bank) | Construction Group 3 | Regulated Segment (North Bank) |
|---|---|---|---|
| CS1 | Section 1 # of Matan Hongshui He Bridge | TK1 | Section 1 # of Matan Hongshui He Bridge |
| CS2 | Section 5 # of Matan Hongshui He Bridge | TK2 | Section 5 # of Matan Hongshui He Bridge |
| CS3 | Section 6 # of Matan Hongshui He Bridge | TK3 | Section 6 # of Matan Hongshui He Bridge |
| CS4 | Section 1 # of Pingnan Third Bridge | TK4 | Section 1 # of Pingnan Third Bridge |
| CS5 | Section 5 # of Pingnan Third Bridge | TK5 | Section 5 # of Pingnan Third Bridge |

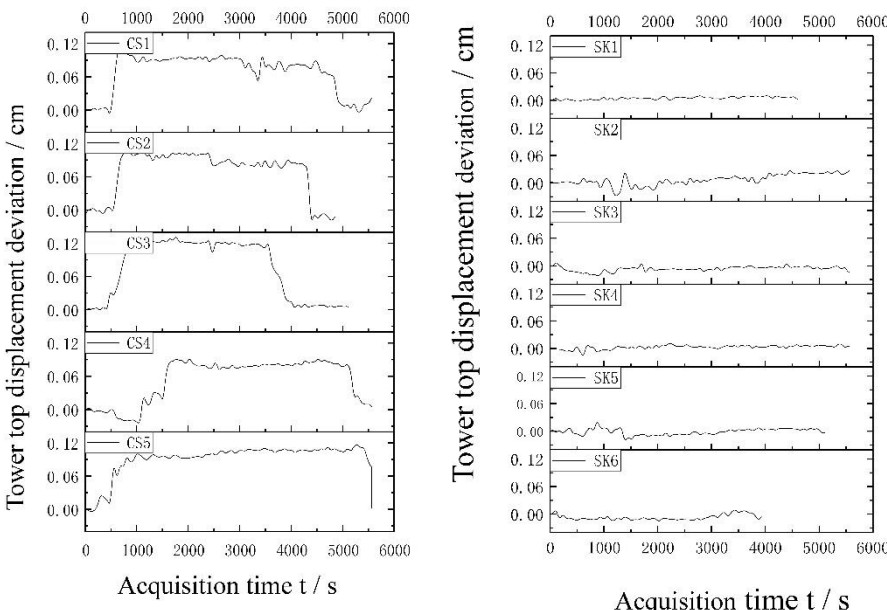

**Figure 17.** The measured results of tower top deviation under normal working conditions and active control of different lifting sections.

Through an active control technology, the finite element results and measured results are within 2 cm, and the displacement of the tower top fluctuates within the range of 8~14 cm in the normal lifting process of the arch ribs. When the maximum value that occurred in Section 6 locates upstream of the right of the south bank, the horizontal deviation of the tower top (the direction toward the river is positive) reaches 14.0 cm, which affect the safety of the tower itself greatly. The mean value of the maximum horizontal deviation of the tower is effectively controlled within 2 cm through active control. The elevation change of arch rib affected by deviation is only 1 mm, greatly reducing the deviation value. The control is quite effective and the experiment objective of smart active control of the horizontal displacement of the tower is basically achieved. Furthermore, the filtered curve includes the displacement of the entire tower from the lifting of the arch ribs to the lifting point in the lifting process, further proving the reliability and accuracy of the GNSS and the good effect of the smart tower deviation control system.

## 6. Conclusions

1.  Based on the study on the tower deviation error formation mechanism in the process of cable lifting, the related formulas of the arch rib elevation change caused by the change of tower as the temporary structure state is deduced. The studies show that the arch rib hanging on it has a linkage effect due to tower stress deformation, and greatly influences the precision of arch girder suspension.
2.  Traditionally, the tower deviation is controlled by arranging a large number of steel strands and manually pulling the steel strands, which wastes a lot of steel strand materials and labor. Now, the cost is greatly reduced by the technology proposed in this paper.
3.  Under normal operation, the influence of active regulation on the stress of each component of the structure is not significant. This means that active regulation has an obvious influence on the lateral displacement of the tower, but its sensitivity to the internal force and stress of the tower is very low, and the tower is in a safe state.
4.  The technology, having been applied to world-class arch bridges, can actively and timely apply pressure to the tower top, correct the deviation of tower in the process of cable lifting, reduce the deviation value, and finally reduce the impact on the precision of the arch girder suspension. Furthermore, it can realize the real-time control of the tower maximum deviation within the scope of ±2 cm. The research results explore and improve the CFST arch bridge construction technology, and provide a construction reference for the tower deviation control of the same types of bridges and more long-span CFST arch bridge with intergraded main cable and buckle. With the development of positioning technology, the measurement accuracy of tower offset will be higher, and this technology will be increasingly widely used.

**Author Contributions:** Conceptualization, N.D. and M.Y.; methodology, M.Y.; data analysis X.Y.; writing—original draft preparation, M.Y and X.Y.; writing—review and editing, M.Y. and X.Y. All authors have read and agreed to the published version of the manuscript.

**Funding:** This research was funded by Nanning City "Yongjiang Project" funded project (2018-01-04), National Natural Science Foundation of China (grant number 51868006), and Nanning Excellent Young Scientist Program (grant number RC20180108).

**Institutional Review Board Statement:** Not applicable.

**Informed Consent Statement:** Not applicable.

**Data Availability Statement:** The testing and analysis data used to support the findings of this study are included within the article.

**Conflicts of Interest:** The authors declare no conflict of interest.

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
