# Peer review of "Intelligent Active Correction Technology and Application of Tower Displacement in Arch Bridge Cable Lifting Construction"

_applsci, doi:10.3390/app11219808_

Round 1

Reviewer 1 Report

Dear author,
Please consider the following remarks.
1.    It is better to explain the abbreviation CFST as it has done in the article's text.
2.    The Abstract requires revision. Thе Abstract should emphasize the research results. The following structure of the Abstract is recommended:
•    definition of the research object;
•    descriptions of the research method;
•    research results.
3.    The INTRODUCTION section does not properly motivate the research by analyzing modern publications in leading international journals. The section is too short. Only twelve cited publications are in English. Consequently, the author's research is not motivated. 
4.    The review is not like a search for a ready-made solution to the problem facing the author. The absence of such a solution in publications has not been formulated. The object of research, the goal, and the study's objectives are not clearly defined at the end of the INTRODUCTION section.
5.    The manuscript looks more like a technical report than a research paper. The structure of the article does not comply with the IMRAD format. Please rename the sections of the article according to the IMRAD format. It is recommended to use non-IMRAD titles only for subsections, paragraphs, subparagraphs (with their multi-level numbering).
6.    There is no clear description of the methods by which the author obtains new scientific results in the PRINCIPLE AND METHOD section. 
7.    The author applies "the simplified model" (Line 84). However, there is no list of simplifications and assumptions on which this simplified model operates. There is no analysis of the applicability of simplifications and assumptions. Thus, the absence of a list casts doubt on all the theoretical results of the author.
8.    The article is in many ways reminiscent of a. This case study stile is partly due to the lack of a classification of bridges and the lack of clarification for which type of bridges the author's results are valid.
9.    The CONCLUSIONS section comes from the results, but it is still a simple summary of the values obtained. The academic evaluation is lacking.

Author Response

I sincerely thank the reviewers and the editorial department for their valuable comments. According to the suggestions of reviewers and the editorial department, the author has revised and improved the paper. The specific instructions are as follows:

Reviewers' opinions

Q1. It is better to explain the abbreviation CFST as it has done in the article's text.

A1. Thank you reviewers for their valuable comments. According to the recommendations of reviewers, CFST is explained (See the highlighted content in yellow from lines 19to 20 of the revised manuscript)

Q2. The Abstract requires revision. Thе Abstract should emphasize the research results. The following structure of the Abstract is recommended:

  • definition of the research object;
  • descriptions of the research method;
  • research results.

A2. Thank you reviewers for their valuable comments. According to the recommendations of reviewers, Relevant modifications have been made to the summary .(See the highlighted content in yellow from lines 11to27)

Q3. The INTRODUCTION section does not properly motivate the research by analyzing modern publications in leading international journals. The section is too short. Only twelve cited publications are in English. Consequently, the author's research is not motivated.

A3. Thank you reviewers for their valuable comments. According to the recommendations of reviewers, Relevant modifications have been made to the introduction (See the highlighted content in yellow from lines 31to 70 of the revised manuscript).

Q4. The review is not like a search for a ready-made solution to the problem facing the author. The absence of such a solution in publications has not been formulated. The object of research, the goal, and the study's objectives are not clearly defined at the end of the INTRODUCTION section.

A4. Thank you reviewers for their valuable comments. According to the recommendations of reviewers, Relevant modifications have been made to the introduction (See the highlighted content in yellow from lines 31to 70 of the revised manuscript).

Q5. The manuscript looks more like a technical report than a research paper. The structure of the article does not comply with the IMRAD format. Please rename the sections of the article according to the IMRAD format. It is recommended to use non-IMRAD titles only for subsections, paragraphs, subparagraphs (with their multi-level numbering).

A5. Thank you reviewers for their valuable comments. According to the recommendations of reviewers, Relevant modifications have been made according to the modification opinions.

Q6. There is no clear description of the methods by which the author obtains new scientific results in the PRINCIPLE AND METHOD section.

A6. Thank you reviewers for their valuable comments. According to the recommendations of reviewers, The steps of monitoring technology are introduced.(See the highlighted content in yellow from lines 59to70 and lines 234to240 of the revised manuscript).

Q7. The author applies "the simplified model" (Line 84). However, there is no list of simplifications and assumptions on which this simplified model operates. There is no analysis of the applicability of simplifications and assumptions. Thus, the absence of a list casts doubt on all the theoretical results of the author.

A7. Thank you reviewers for their valuable comments. According to the recommendations of reviewers, In order to explore the basic principle of tower deflection, the following two assump-tions are made.(See the highlighted content in yellow from lines 78to83 of the revised manuscript).

Q8. The article is in many ways reminiscent of a. This case study stile is partly due to the lack of a classification of bridges and the lack of clarification for which type of bridges the author's results are valid.

A8. Thank you reviewers for their valuable comments. According to the recommendations of reviewers, This case is applicable to CFST(concrete filled steel tubular arch bridge )(See the highlighted content in yellow from lines 412o421 of the revised manuscript).

Q9. The CONCLUSIONS section comes from the results, but it is still a simple summary of the values obtained. The academic evaluation is lacking.

A9. Thank you reviewers for their valuable comments. According to the recommendations of reviewers, The advantages of this method are explained .(See the highlighted content in yellow from lines 404to407 of the revised manuscript).

Reviewer 2 Report

The reviewer appreciates the experimental and numerical work done by the authors. The technical contents of the paper are in general interesting. The technology adopted for cable-arch bridges is promising. The corresponding findings are useful for the bridge engineering community, and for improving future structural health monitoring systems. Nevertheless, I do not recommend the publication of the manuscript in the “Applied Sciences’ unless following major modifications are done in the article:

1) The current state of knowledge relating to the manuscript topic is not covered and clearly presented, and the authors’ contribution and novelty are not emphasized. In this regard, the authors should make their effort to address these issues, by adding additional comments on the state of the art and the proposed aspects.

2) Introduction and Conclusions should be improved to sharply emphasize the contribution of the existing methodologies and clearly trace further developments.

3) The article could be improved by inserting a table where the literature is discussed to briefly and schematically characterize each reference (e.g. authors, year, lab experiments, topic, and findings). This could help to give more strength and significance to the state-of-the-art article. And, moreover, by introducing original figures with schemes to explain the driving ideas traced by the literature review.

4) Objectives and findings should be presented more clearly (e.g., using the following division of the Sections: Introduction, Review of Existing Methods, Analytical Model and Formulas, Proposed Methodology of Monitoring, Application in Cable-Arch Bridges, Finite Element (FE) modeling, Comparison between FE and Experimental Results, Discussion and Conclusions). The current main sections appear not very well organized and divided.

5) Additional comments should be added in regard to the practical value of this research, i.e., how the industry can profit from it.

6) The steps of the proposed monitoring technology (how it works in the field) are not presented in detail. Please, review the corresponding parts.

7) Compared with other monitoring techniques, the superiority of the proposed one is not clearly explained. Please, review the corresponding parts.

8) In a cable-arch bridge or in a cable-stayed bridge, the flexural rigidity of slender cables can generally be neglected. Conversely, it cannot be negligible for the shortest ones. Please, specify this feature properly within the Sections which treat analytical and numerical modeling. With regards, please, also refer and cite the following literature:
-  Estimate of the axial force in slender beams with unknown boundary conditions using one flexural mode shape, J. Sound Vib. 332 (18) (2013) 4122–4135.
-  A novel tension estimation approach for elastic cables by elimination of complex boundary condition effects employing mode shape functions, Eng. Struct. 166 (2018) 152–166.
-  Bending tests for the structural safety assessment of space truss members, Int. J. Space Struct. 33 (3–4) (2018) 138–149.
-  Tension determination for suspenders of arch bridge based on multiple vibration measurements concentrated at one end, Measurement 123 (2018) 254–269.

9) In the Reviewer’s opinion, the first-order critical coefficient is unclear. What is this parameter exactly ? Please, specify it better within the text.

10) I suggest to the authors to edit all the text of the article with the help of a native English speaker who should be an expert in the domain of “Structural Engineering”. Grammar, punctuation, spelling, verb usage, sentence structure, conciseness, readability and writing style can be improved.

Author Response

I sincerely thank the reviewers and the editorial department for their valuable comments. According to the suggestions of reviewers and the editorial department, the author has revised and improved the paper. The specific instructions are as follows:

Reviewers' opinions

Q1. The current state of knowledge relating to the manuscript topic is not covered and clearly presented, and the authors’ contribution and novelty are not emphasized. In this regard, the authors should make their effort to address these issues, by adding additional comments on the state of the art and the proposed aspects.

A1. Thank you reviewers for their valuable comments. According to the recommendations of reviewers, The research provides the scientific and technological problems to be solved and the author's contribution (See the highlighted content in yellow from lines 59to 70 of the revised manuscript)

Q2. Introduction and Conclusions should be improved to sharply emphasize the contribution of the existing methodologies and clearly trace further developments.

A2. Thank you reviewers for their valuable comments. According to the recommendations of reviewers, Relevant modifications have been made .(See the highlighted content in yellow from lines 11to27 and lines 404to422 of the revised manuscript)

Q3. The article could be improved by inserting a table where the literature is discussed to briefly and schematically characterize each reference (e.g. authors, year, lab experiments, topic, and findings). This could help to give more strength and significance to the state-of-the-art article. And, moreover, by introducing original figures with schemes to explain the driving ideas traced by the literature review.

A3. Thank you reviewers for their valuable comments. According to the recommendations of reviewers, The corresponding literature is discussed in the table 1 .(See the highlighted content in yellow from lines 58to 70 of the revised manuscript).

Q4. Objectives and findings should be presented more clearly (e.g., using the following division of the Sections: Introduction, Review of Existing Methods, Analytical Model and Formulas, Proposed Methodology of Monitoring, Finite Element (FE) modeling, Comparison between FE and Experimental Results, Discussion and Conclusions). The current main sections appear not very well organized and divided.

A4. Thank you reviewers for their valuable comments. According to the recommendations of reviewers, Relevant modifications have been made according to the modification opinions.

Q5. Additional comments should be added in regard to the practical value of this research, i.e., how the industry can profit from it.

A5. Thank you reviewers for their valuable comments. According to the recommendations of reviewers, The practical value has been described .(See the highlighted content in yellow from lines 404to407 of the revised manuscript).

Q6. The steps of the proposed monitoring technology (how it works in the field) are not presented in detail. Please, review the corresponding parts.

A6. Thank you reviewers for their valuable comments. According to the recommendations of reviewers, The steps of monitoring technology are introduced.(See the highlighted content in yellow from lines 234to240 of the revised manuscript).

Q7. Compared with other monitoring techniques, the superiority of the proposed one is not clearly explained. Please, review the corresponding parts.

A7. Thank you reviewers for their valuable comments. According to the recommendations of reviewers, The advantages of this method are explained .(See the highlighted content in yellow from lines 404to407 of the revised manuscript).

Q8. In a cable-arch bridge or in a cable-stayed bridge, the flexural rigidity of slender cables can generally be neglected. Conversely, it cannot be negligible for the shortest ones. Please, specify this feature properly within the Sections which treat analytical and numerical modeling. With regards, please, also refer and cite the following literature:

A8. Thank you reviewers for their valuable comments. According to the recommendations of reviewers, Relevant literature has been consulted (See the highlighted content in yellow from lines 264to266 of the revised manuscript).

Q9. In the Reviewer’s opinion, the first-order critical coefficient is unclear. What is this parameter exactly ? Please, specify it better within the text.

A9. Thank you reviewers for their valuable comments. the first-order critical coefficient is Stability coefficient. (See the highlighted content in yellow from lines 318to326 of the revised manuscript).

Q10. I suggest to the authors to edit all the text of the article with the help of a native English speaker who should be an expert in the domain of “Structural Engineering”. Grammar, punctuation, spelling, verb usage, sentence structure, conciseness, readability and writing style can be improved.

A10. Thank you reviewers for their valuable comments. According to the recommendations of reviewers, This article revised the grammar related issues .

Round 2

Reviewer 1 Report

Dear author,
There are two chapters with the same title:
2. Analytical Model and Formulas
3. Analytical Model and Formulas.
So I could not analyze this manuscript.

Author Response

I sincerely thank the reviewers and the editorial department for their valuable comments. According to the suggestions of reviewers and the editorial department, the author has revised and improved the paper. The specific instructions are as follows:The section title has been modified

Reviewer 2 Report

The reviewer appreciates the review done by the authors. Nevertheless, I do not recommend the publication of the manuscript in the “Applied Sciences’ unless following modifications are done in the article:

- In a cable-arch bridge or in a cable-stayed bridge, the flexural rigidity of slender cables can generally be neglected. Conversely, it cannot be negligible for the shortest ones. Please, specify this feature properly within the Sections which treat analytical and numerical modeling. With regards, please, also refer and cite the following literature:
a)  Estimate of the axial force in slender beams with unknown boundary conditions using one flexural mode shape, J. Sound Vib. 332 (18) (2013) 4122–4135.
b)  A novel tension estimation approach for elastic cables by elimination of complex boundary condition effects employing mode shape functions, Eng. Struct. 166 (2018) 152–166.
c)  Bending tests for the structural safety assessment of space truss members, Int. J. Space Struct. 33 (3–4) (2018) 138–149.

Author Response

I sincerely thank the reviewers and the editorial department for their valuable comments. According to the suggestions of reviewers and the editorial department, the author has revised and improved the paper. The specific instructions are as follows:The section title has been modified:

  In a cable-arch bridge or in a cable-stayed bridge, the flexural rigidity of slender cables can generally be neglected. Conversely, it cannot be negligible for the shortest ones.The accuracy of cable force calculation is directly related to the boundary conditions and calculation model at both ends of the cable. Stiffness and boundary conditions are the two most important factors in the establishment of short cable model. The accurate measurement method of cable force is adopted[32], which comprehensively considers the influence of bending stiffness and boundary conditions. At the same time, the bending stiffness is taken as an implicit calculation parameter, which avoids the problem that it is difficult to identify accurately, reduces the measurement difficulty and improves the measurement accuracy. [33-35]
